

# Marked variations in gut microbiota and some innate immune responses of fresh water crayfish, marron (*Cherax cainii*, Austin 2002) fed dietary supplementation of *Clostridium butyricum*

Md Javed Foysal[1,2], Thi Thu Thuy Nguyen[1], Md Reaz Chaklader[1], Muhammad A.B. Siddik[1,3], Chin-Yen Tay[4], Ravi Fotedar[1] and Sanjay Kumar Gupta[5]

[1] School of Molecular and Life Sciences, Curtin University, Bentley, Western Australia, Australia
[2] Department of Genetic Engineering and Biotechnology, Shahjalal University of Science & Technology, Sylhet, Bangladesh
[3] Department of Fisheries Biology and Genetics, Patuakhali Science and Technology University, Patuakhali, Bangladesh
[4] Helicobacter Research Laboratory, Marshall Centre for Infectious Disease Research and Training, School of Biomedical Sciences, University of Western Australia, Perth, Western Australia, Australia
[5] ICAR-Indian Institute of Agricultural Biotechnology, Ranchi, Jharkhand, India

Corresponding author
Md Javed Foysal,
mdjaved.foysal@postgrad.curtin.edu.au

## ABSTRACT

This study aimed to investigate the effects of *Clostridium butyricum* as a dietary probiotic supplement in fishmeal based diet on growth, gut microbiota and immune performance of marron (*Cherax cainii*). Marron were randomly distributed into two different treatment groups, control and probiotic fed group. After 42 days of feeding trial, the results revealed a significant ($P < 0.05$) increase in growth due to increase in number of moults in marron fed probiotics. The probiotic diet also significantly enhanced the total haemocyte counts (THC), lysozyme activity in the haemolymph and protein content of the tail muscle in marron. Compared to control, the 16S rRNA sequences data demonstrated an enrichment of bacterial diversity in the probiotic fed marron where significant increase of *Clostridium* abundance was observed. The abundance for crayfish pathogen *Vibrio* and *Aeromonas* were found to be significantly reduced post feeding with probiotic diet. Predicted metabolic pathway revealed an increased activity for the metabolism and absorption of carbohydrate, degradation of amino acid, fatty acid and toxic compounds, and biosynthesis of secondary metabolites. *C. butyricum* supplementation also significantly modulated the expression level of immune-responsive genes of marron post challenged with *Vibrio mimicus*. The overall results suggest that *C. butyricum* could be used as dietary probiotic supplement in marron aquaculture.

## INTRODUCTION

Crayfish harbour complex bacterial communities in the intestine that stimulates various host functions like digestion, absorption, immunity and disease resistance (*Skelton et al., 2017*). Augmentation in the growth and immunity of fish and crayfish with dietary probiotic supplements has long been studied (*Ambas, Fotedar & Buller, 2017*; *Ambas, Suriawan & Fotedar, 2013*; *Didinen et al., 2016*; *Panigrahi et al., 2007*). Probiotics are live microorganisms that can improve the digestive function by influencing the growth of beneficial microorganisms in the gut when administered in adequate amounts (*Irianto & Austin, 2002*; *Dawood et al., 2016*). Probiotic bacteria can improve the growth, immune response, feed utilization and stress response of fish (*Kesarcodi-Watson et al., 2008*; *Iwashita et al., 2015*; *Dawood et al., 2016*) in the host species. Among the bacterial supplements, majority of the studies focussed on the beneficial effects of lactic acid bacteria (LAB), especially *Lactobacillus* species on growth, immunity and disease resistance of fish and crayfish (*Pirarat et al., 2006*; *Didinen et al., 2016*; *Zheng et al., 2017*). *Clostridium butyricum*, a short-chain butyric acid producing bacteria that resides in healthy intestinal flora, has been used as diet supplements to enhance the growth and immune response of fish and broiler chicken (*Song et al., 2006*; *Gao et al., 2013*; *Zhang et al., 2014*; *Ramírez et al., 2017*). In addition, *C. butyricum* as probiotic supplement demonstrated to have inhibitory effects against the colonization of pathogenic bacteria including species from *Vibrio* and *Aeromonas* in the fish gut (*Pan et al., 2008*; *Gao et al., 2013*). However, the impacts of *C. butyricum* on crayfish, especially on marron (*Cherax cainii*) growth performance, gut microbiota and immune gene expression has yet to be explored.

Marron is the third largest freshwater crayfish and iconic to Western Australia (WA). The global demands of marron is very high due to its large harvest size (up to 2 kg), distinctive flavour, disease-free status, high consumer preference, and ability of live transport (*Ambas, Suriawan & Fotedar, 2013*). Slow growth of marron has long been a bottleneck in the expansion of marron aquaculture in WA (*Lawrence, 2007*; *Alonso, 2010*; *Australian Department of Fisheries, 2017*). Although there has not been any current report of infection in marron, future expansion of marron industry could bring this threat especially infections by *Vibrio* species (*Eaves & Ketterer, 1994*; *Bean et al., 1998*; *Sherry et al., 2016*). Therefore, enhancing growth parameters and minimizing the possible incidences of *Vibrio* infections are the two utmost challenges in the expansion of marron aquaculture.

Integration of traditional growth performance analysis methods with recently developed modern technologies like 16S rRNA based high throughput sequencing and bioinformatics pipelines have enabled more comprehensive analysis of feeding effects at a cellular and molecular level of fish (*Allali et al., 2017*; *Miao et al., 2018*). In addition, quantitative real-time polymerase chain reaction (PCR) has been widely used to measure the relative expression level of immune responsive genes for fish (*Mahanty et al., 2017*). The present investigation was designed to evaluate the effects of dietary supplementation of *C. butyricum* on health, gut microbial community and immune related gene expression of marron.

## MATERIALS AND METHODS

### Experiment set-up

A total of 24 marron with a mean weight of 69.65 ± 1.04 g were transported alive from Blue Ridge Marron, Manjimup, (34.2019°S, 116.0170°E) to Curtin Aquatic Research Laboratory (CARL), Bentley campus, Curtin University, Western Australia. Marron were then randomly distributed into six different tanks (150 L capacity) and acclimatized for two weeks before starting the feeding trial. During acclimatization, marron were fed with commercial basal diet named marron pellet (Glenn Forest, Perth, Australia), composed of 28% crude protein, 9% crude fat, 8.5% moisture, and 5% crude ash. After acclimatization, marron were disseminated into two distinct treatment groups' viz. control and probiotic fed group with a stocking density of four marron per tank. Constant aeration to each tank was distributed by air stones (Aqua One, Perth, Australia) and fixed temperature of 22 ± 0.5 °C was maintained using submersible thermostat set (Aqua One, Perth, Australia). Each marron were individually nurtured in a special cage prepared of plastic mesh (0.8–8.0 mm thickness) to avoid cannibalism as described previously (*Ambas, Suriawan & Fotedar, 2013*). Approximately 30% of water was exchanged every day from each tank to remove uneaten feed and faecal debris by siphoning.

### Probiotic feed formulation

The probiotic feed was formulated at CARL according to the procedure used by *Ambas, Suriawan & Fotedar (2013)*. The fishmeal based basal diet was used for probiotic feed formulation (Table 1). The ingredients of the commercial basal feed were gently passed through 100 μm mesh sieve and homogenized to get uniform particle size. *C. butyricum* purchased from Advanced Orthomolecular Research (AOR, Calgary, Canada) was cultured on Clostridial Agar (Sigma-Aldrich, MO, USA) followed by sub-culture on Clostridial agar. Then a serial dilution and subsequent plate counts were performed to obtain the desire bacterial concentration. The bacteria (*C. butyricum*) were then added at $10^7$ CFU/mL per kg of feed as described in previous studies (*Ramírez et al., 2017*) and mixed uniformly. Pellet were prepared with mince mixture followed by vacuum drying oven at 37 °C for overnight and storage at 4 °C until used. Final proximate compositions and bacterial concentration of diets were determined as per the method of Association of Official Analytical Chemists, AOAC (*AOAC, 2006*) and plate counts on Clostridial agar (Table 1). During experimental trial, each marron were fed once every day at 5 PM for 42 days at a rate of 1.5% of total marron biomass per tank (*Ambas, Fotedar & Buller, 2017*). Control group fed basal diet and probiotic group served *C. butyricum* supplemented diet.

### Monitoring of water quality parameters

The temperature and pH of experimental tank water was measured using portable waterproof °C/mV/pH meter (CyberScan pH 300; Eutech Instruments, Singapore). The levels of dissolved oxygen (DO) in water was monitored by digital DO meter (YSI55; Perth Scientific, Australia). The concentrations of nitrate ($NO_3^-$) and nitrite ($NO_2^-$) were checked using Hach DR/890 Colorimeter (Hach, Loveland, CO, USA) following the method of cadmium reduction and diazotization (*Hoang et al., 2016*). Phosphate ($PO_4^-$)

**Table 1  Feed ingredients and proximate composition (% dry weight) of the diet used in this study.**

| Ingredients[a] | Basal diet |
|---|---|
| Fishmeal | 41 |
| Soya bean meal | 10 |
| Wheat | 37 |
| Corn starch | 4.80 |
| Cod liver oil | 4.20 |
| $CaCO_3$ | 0.02 |
| Vitamin premix | 0.23 |
| Vitamin C | 0.05 |
| Cholesterol | 0.50 |
| Lecithin-Soy | 1 |
| Betacaine | 1.20 |
| Total | 100 |
| [b]CP% | 29.93 |
| [b]Lipid % | 7.12 |
| [b]GE MJ $kg^{-1}$ | 18.21 |
| [b]*C. butyricum* CFU/mL | $1.01 \times 10^7$ |

Notes.

[a]All ingredients were procured and feeds were formulated by Glen Forest Specialty Feeds, Western Australia.

Abbreviations: CP, crude protein; GE, gross energy; MJ, Mega joule.

[b]Final proximate composition of the experimental diet.

level in water was examined by ascorbic acid standard 4500-PE method, as described by Huong (*Mai, Fotedar & Fewtrell, 2010*). The concentrations of ammonia ($NH_3$) in water was monitored using ammonia test kit (Hach, Loveland, CO, USA).

## Marron sampling

For analysis of haemolymph parameters (lysozyme, haemolymph osmolality and total haemocyte count) health indices (protein and energy in tail muscle), one randomly selected marron from each tank ($N = 6$) was selected. However, for DNA extraction and microbiome analysis, two randomly selected marron from each tank ($N = 12$) were scarified followed by careful separation of hindgut. The hindgut contents of two samples from each tank were homogenized and pooled together ($N = 6$), and then transferred into 1.5 mL Eppendorf. Finally, for immune gene expression, the whole intestine tissue sample from one randomly selected marron from each tank ($N = 6$) was used after challenge test. Sample from each marron was used for the haemolymph parameters (before sacrifice), biochemical assay (after sacrifice), and molecular analysis (microbiome and gene expression).

## Analysis of growth and immune parameters

Before starting the experiment and at the end of the trial, the weight of each marron was recorded. The weight gain (WG), specific growth rate (SGR) and feed conversion ratio (FCR) were measured at the end of the experiment according to (*Ambas, Fotedar & Buller, 2017*). The number of moults in each tank was monitored carefully. For analysis of immune parameters like haemolymph osmolality (HO), lysozyme and total haemocyte counts

(THC), the haemolymph was extracted carefully from the pericardial cavity of marron with a 0.5 mL syringe containing 0.1 mL pre-cooled anticoagulant (0.1%) glutaraldehyde in 0.2M sodium cacodylate, (pH 7.0 ± 0.2). The HO of marron was measured using Cryoscopic Osmometer-Osmomet 030 (Gonotec, Berlin, Germany) as described by (*Sang & Fotedar, 2004*). THC of the extracted haemolymph from each replicate of two treatment group was calculated under a hemocytometer (Nauabuer, Germany) with 100X magnification according to method described by *Ambas, Fotedar & Buller (2017)*. The lysozyme activity of marron haemolymph in control and probiotic fed groups was measured using turbidimetric assay (*Mai & Fotedar, 2018*; *Foysal et al., 2019*). Briefly, 50 µL of hemolymph samples (anticoagulant added) were disseminated into 96-well microtiter-plate (Iwaki, Tokyo, Japan). After incubation for 15 min at room temperature, 50 µL of PBS (0.25 mg/mL) suspended *Micrococcus lysodeiktikus* (Sigma-Aldrich, St. Louis, MO, USA) solution was added into separate wells of the same plate. The absorbance of each well in the microtiter-plate was then monitored in a MS212 reader (Titertek Plus, Tecan, Grodig, Austria) at 450 nm wavelength according to manufacturer's instructions. The assay was based on the rate of *Micrococcus lysodeiktikus* cell lysis in terms of absorbance change caused by lysozyme in the hemolymph. The measurements were taken at every 2 min intervals for 20 min (U/mL).

## Analysis of biochemical composition

The protein and energy content in the tail muscle and fat content in the hepatopancreas were measured according to methods described by the Association of Official Analytical Chemists, AOAC international (*AOAC, 2006*). The percentage of crude protein in oven dried tail muscle was measured following Kjeldahl method (N ×6.25) in Kjeltec Auto 1030 analyser (Foss Tecator, Höganäs, Sweden) (*Jim, Garamumhango & Musara, 2017*). Fat content in hepatopancreas was analysed following Soxhlet ether extraction method using Soxtec System HT6 (Tecator, Höganäs, Sweden) (*Jim, Garamumhango & Musara, 2017*). The amount of total gross energy in the tail muscle was calculated using a bomb calorimeter (IKA, Heitersheim, Germany).

## High throughput sequencing

At the end of the experiment, gut samples prepared as described in marron sampling were used for DNA extraction. Due to special role in digestion, absorption and immunity, hindgut was selected for 16S rRNA sequencing (*Wang et al., 2018*). The bacterial DNA was extracted using DNeasy Blood and Tissue Kit (Qiagen, Crawley, UK) following manufacturer's instructions. Extracted DNA was checked for quantity in a NanoDrop Spectrophotometer (Thermo Fisher Scientific, Waltham, MA, USA) and diluted to 30 ng/µL as final concentration. Thirty cycles of PCR amplification were performed in a BioRad S100 Gradient Thermal Cycler (Bio-Rad Laboratories, Inc., Foster City, CA, USA) with V3–V4 sequencing primers (Part # 15044223 Rev. B) and Hot Start *Taq* 2X Master Mix (BioLab Inc., Lawrenceville, GA, USA). Amplified PCR products were separated by 1% agarose gel and visualized under a gel documentation system (FujiFilm LAS–4000 Image Analyzer, Boston Inc., Foster City, CA, USA). Secondary PCR was applied for the barcoding

**Table 2  Primers used for gene expression analysis in present study.**

| Primer name | Forward sequence (5′–3′) | Reverse sequence (5′–3′) | Reference |
|---|---|---|---|
| proPO | GCCAAGGATCTTTGTGATGTCTT | CGGCCGGCCAGTTCTAT | Liu et al. (2013) |
| cytMnSOD | AGGTCGAGCAAGCAGGTGTAG | GTGGGAATAAACTGCAGCAATCT | Liu et al. (2013) |
| PcCTSL | CGGATCACTGGAGGGTCAAACACTT | GCAATTTTCATCCTCGGCATCAT | Dai et al. (2017) |
| IL-8 | CTATTGTGGTGTTCCTGA | TCTTCACCCAGGGAGCTTC | Miao et al. (2018) |
| IL-10 | CAGTGCAGAAGAGTCGACTGCAAG | CGCTTGAGATCCTGAAATATA | Miao et al. (2018) |
| IL-17F | GTCTCTGTCACCGTGGAC | TGGGCCTCACACAGGTACA | Miao et al. (2018) |
| β-actin | TTGAGCAGGAGATGGGAACCG | AGAGCCTCAGGGCAACGGAAA | Miao et al. (2018) |

Notes.

Abbreviation: IL, Interleukin; PcCTSL, Cathepsin L; proPO, Prophenoloxidase; cytMnSOD, Cytosolic manganese superoxide dismutase.

of 16S rRNA PCR amplicon of each sample according to the Illumina standard protocol (Part # 15044223 Rev. B). The samples were then sequenced up to 20,000 reads on an Illumina MiSeq platforms (Illumina Inc., San Diego, CA, USA) at Harry Perkins Institute of Medical Research, Western Australia, using a v3 kit (600 cycles, Part # MS-102-3003).

## Gene expression analysis in challenged marron

Six genes, namely, interleukin 8, interleukin 10, interleukin 17F, cathepsin L, prophenoloxidase and cytosolic manganese superoxide dismutase (IL-8, IL-10, IL-17F, PcCTSL, proPO and cytMnSOD) that have been reported to be associated with innate immunity of fish and crayfish (Liu et al., 2013; Jiang et al., 2015; Dai et al., 2017; Miao et al., 2018) were selected. The primers used for the selected genes are listed in Table 2. Crayfish pathogen *Vibrio mimicus* was collected from the Department of Food and Agriculture, Western Australia. At the end of the feeding trial, each selected marron was injected with previously prepared $2 \times 10^8$ CFU/mL stock solution of *V. mimicus* (Ambas, Suriawan & Fotedar, 2013). Fifty microliters of bacterial solution were injected through the base of the fifth thoracic leg of marron (Ambas, Suriawan & Fotedar, 2013). Control marron were injected with 50 μL of phosphate buffer saline (PBS). Injected marron were subjected to RNA extraction from intestine tissue after 24 h of bacterial challenge. For RNA extraction, intestine tissue samples from challenged marron were initially stored at −80 °C with *RNA Later* solution (Sigma-Aldrich, Munich, Germany). The samples were thawed, dried, homogenized, ground into fine powder, and finally pooled together according to groups prior to use for RNA extraction. Approximately, 5 mg of pooled tissue sample from each group was used for RNA extraction using RNeasy Mini Kit (Qiagen, Hilden, Germany). During extraction process, RNase free DNase-I (Qiagen, Hilden, Germany) was added according to manufacturer's instructions for removing of DNA related impurities. The quality and quantity of the extracted RNA was checked in 1% gel electrophoresis and NanoDrop spectrophotometer 2000c (Thermo Fisher Scientific, Waltham, MA, USA). The cDNA library was prepared using Omnicript RT kit (Qiagen, Hilden, Germany) from 1 μg of RNA. Real-time PCR amplification was performed using PowerUp™ Cyber Green Master Mix (Thermo Scientific) with 7500 Real-Time PCR System (Applied Biosystems, Foster City, CA, USA) to measure the relative expression level of six selected genes. The

relative expression pattern of each target gene in two different groups was calculated using the $2^{-\Delta\Delta CT}$ method, after normalisation against β-actin reference gene.

## Bioinformatics

Unless any modifications mentioned, the "pipelines" for downstream sequence analysis and bioinformatics were used with default parameters. After extraction, the initial quality of 16S rRNA sequences were checked in fastQC pipelines (*Andrews, 2010*). Sickle program was used for the trimming of quality reads (*Joshi & Fass, 2011*). Following trimming, reads having a length of less than 200 bp were discarded. MeFiT pipeline was used for the joining of overlapping pair-ends reads (*Parikh et al., 2016*). Filtering of chimeric sequences, *de novo* greedy clustering of 16S rRNA sequences into Operational Taxonomic Units (OTUs) at 97% similarity threshold and removal of singleton OTUs were conducted using micca otu (version 1.7.0) (*Albanese et al., 2015*). Taxonomic assignment of the representative OTUs was performed using mica classify against SILVA 132 database clustered at 97% identity (*Quast et al., 2013*). Multiple sequence alignment of the obtained OTUs was performed using PASTA algorithm (*Mirarab et al., 2015*). The rarefaction depth value was set at 4,672 and subsequent calculation of alpha and beta diversities was performed using QIIME pipeline (version 1.9.1) (*Kuczynski et al., 2012*). Briefly, the alpha diversity was calculated based on observed species and diversity indices (Shannon, Simpson, Alpha Fisher and Chao1) using vegan package in Rstudio (*Oksanen et al., 2018*). Non-parametric *t*-test for two samples was performed to compare the alpha diversity metrics between the control and probiotic-fed samples. Principle coordinates analysis (PCoA) was performed to visualize separation of samples using Bray-Curtis and jaccard abundance metrics. Non-parametric statistical test of the distance metric was done using ANOSIM (1,000 permutations). PCoA plots were generated using PhyloToAST software (*Dabdoub et al., 2016*). The assigned bacterial genus from two different feeding groups were plotted in Venn diagram, generated using FunRich tool (*Benito-Martin & Peinado, 2015*). Indicator bacterial genus in two different conditions was differentiated from rarefy 16S rRNA sequence data using Linear Discriminant Analysis Effect Size, LEfSe (University of Auckland, New Zealand) (*Segata et al., 2011*). For predicting the metabolic pathways through 16S rRNA sequences of two different fed groups, Piphillin pipeline (http://secondgenome.com/Piphillin) was applied with supports of KEGG (May, 2017 release), BioCyc 21.0 databases and LEfSe algorithm (*Segata et al., 2011*; *Iwai et al., 2016*). One way analysis of variance (ANOVA) was used to calculate any significant differences ($P < 0.05$) among the numerical data including growth (WG, SGR, FCR), biochemical compositions (protein, fat, energy), lysozyme, haemolymph osmolality and THC obtained from two different feeding conditions.

## Calculations

Following calculations were used:

Weight gain, WG = [(mean final body weight − mean initial body weight)]/[mean initial body weight] (*Ramírez et al., 2017*)

Specific growth rate, SGR (% day) = [(ln mean final body weight − ln mean initial body weight)/number of days] ×100 (*Ambas, Fotedar & Buller, 2017*)

**Table 3  Mean ± SE of some health parameters of marron after feeding trials.**

| Parameters | Control | Probiotic |
|---|---|---|
| WG | 21.66 ± 1.78 | 26.24 ± 0.81[a] |
| SGR | 0.45 ± 0.034 | 0.53 ± 0.031[a] |
| Protein[b] | 84.4 ± 0.57 | 88.7 ± 0.83[a] |
| Energy[b] | 20038.8 ± 56.06 | 20241.1 ± 88.25 |
| Fat[c] | 8.8 ± 0.09 | 8.4 ± 0.06 |
| Lysozyme | 0.42 ± 0.005 | 0.48 ± 0.009[a] |
| THC | 8.2 ± 0.09 | 9.5 ± 0.21[a] |
| HO | 0.41 ± 0.004 | 0.44 ± 0.011 |

Notes.

Abbreviations: WG, weight gain; SGR, specific growth rate; HM, hepatopancreas moisture; THC, total haemocyte count; HO, haemolymph osmolality.

[a]Significantly different at $\alpha$-level of 0.05.

[b]From tail muscle.

[c]From hepatopancreas.

Feed conversion ratio, FCR = feed intake (dry weight)/weight gain (wet g) (*Ramírez et al., 2017*)

Haemolymph osmolality, HO = 3× osmolality of the mix − 2× osmolality of anticoagulant (*Sang & Fotedar, 2004*)

Total haemocyte counts, THC = (cells counted x dilution factor × 1,000)/volume of grid (0.1 mm$^3$) (*Ambas, Fotedar & Buller, 2017*).

# RESULTS

## Water quality parameters

The water quality parameters were recorded within the normal range for optimum growth of marron according to *Nugroho & Fotedar (2013)*. Temperature was maintained at $22.3 \pm 0.014 - 22.4 \pm 0.01$ °C, pH was recorded within the range of $7.5 \pm 0.01 - 7.6 \pm 0.01$, dissolved oxygen levels between $6.51 \pm 0.01$ and $6.65 \pm 0.01$, nitrate and nitrite concentration of $0.046 \pm 0.01 - 0.049 \pm 0.01$ and $0.015 \pm 0.01 - 0.018 \pm 0.01$ were recorded, phosphate concentration was found to be varied within the range of $0.36 \pm 0.01 - 0.52 \pm 0.02$, and ammonia concentration of $0.20 \pm 0.01 - 0.22 \pm 0.01$ was observed in water during the experiment.

## Probiotic supplement enhanced the growth and haemolymph parameters

After 42 days, the marron in the probiotic fed tanks gained significant ($P < 0.05$) weight gain (WG) and specific growth rate (SGR) than the control (Table 3). A total of 11 marron were moulted in the probiotic fed tanks compared to three in the control tanks, shown significant ($P < 0.05$) increase of moult number in probiotic fed marron. Multiple regression analysis revealed a strong "Pearson" correlation (0.99) between the number of moults and weight gain (Fig. 1A). PCoA plot also revealed a complete separation of probiotic fed group based on moulting numbers and weight gain (Fig. 1B). Probiotic diet also exhibited significant ($P < 0.05$) impacts on two out of three haemolymph parameters, total haemocyte counts

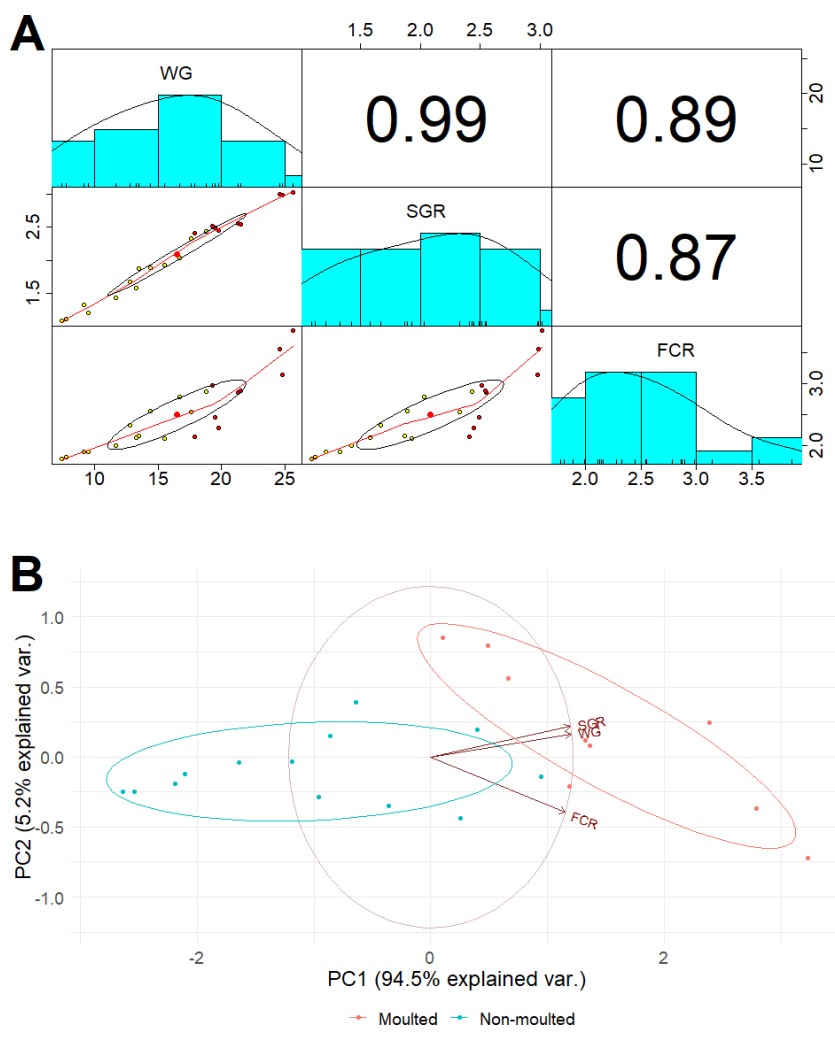

**Figure 1** (A) Multiple regression analysis showing the relationship between the number of moults and health indices of marron after trial; (B) PCoA plot displaying the separation of marron regarding moult counts in two different feeding conditions.

(THC) and lysozyme. Among two tail muscle parameters, significant ($P < 0.05$) increase in protein content was observed in probiotic fed marron while the variations were insignificant ($P > 0.05$) for total gross energy, and fat in the hepatopancreas in the marron fed probiotic feed (Table 2).

## Probiotic diet modulated the microbial communities in the distal gut

The probiotic supplemented diet exhibited a distinct effect on gut microbial community of marron after feeding trial (Fig. 2). The 16S rRNA sequence from eight pooled hindgut samples generated 128,567 high quality reads after quality trimming and removing singletons, that were classified into 83 OTUs and 49 genus. Besides 18 shared genus in both groups (control and probiotic), the probiotic feed group had an additional 31 unique genus compared to four in control diet group, indicating higher species diversity

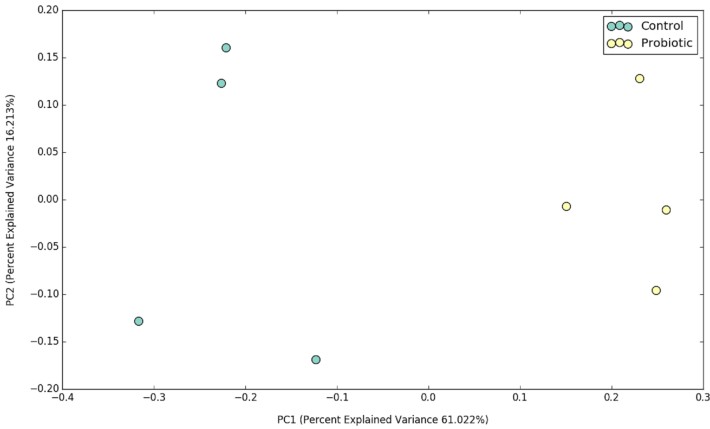

**Figure 2**   PCoA plot representing the impacts of two different feeds on gut microbial communities of marron after 42 days of feeding trial.

**Table 4**   Diversity index (Mean SE) of bacterial genus in marron gut after feeding trials.

| Condition | OTUs (SE) | Shannon (SE) | Simpson (SE) | Fisher alpha (SE) | Chao1 (SE) |
|---|---|---|---|---|---|
| Control | 15.8 (1.9) | 1.51 (0.1) | 0.69 (0.1) | 7.59 (0.2) | 86.74 (5.8) |
| Probiotic | 30.3 (11.5) | 3.19 (0.1)[c] | 0.82 (0.1)[b] | 16.7 (2.1)[a] | 128.48 (8.6)[a] |

**Notes.**
[*]Significantly different at $\alpha$-level of 0.05.
[**]Significantly different at $\alpha$-level of 0.005.
[c]Significantly different at $\alpha$-level of 0.001.

(Fig. 3). At phylum level, control diet group dominated by the *Proteobacteria* (96.5%) while *Fusobacteria* (53.4%) and *Proteobacteria* (44.1%) shared 97.5% of OTUs in the probiotic fed group (Fig. 4A). At genus level, control group demonstrated more abundance for *Aeromonas* (47.1%) and *Vibrio* (31.8%) whereas in probiotic fed group, *Hypnocyclicus* (57.7%) was the most dominant bacteria, followed by *Vibrio* (13.1%) and *Aeromonas* (9.7%). A significant reduction of *Vibrio* (13.1%) and *Aeromonas* (9.7%) counts was noticed in the probiotic fed group at the end of the trial (Fig. 4B). Although the values for major diversity indices such as Shannon, Simpson, Fisher Alpha and Chao1 were significantly ($P < 0.05$) higher but more pronounced effects was observed only for Shannon ($P < 0.001$) and Simpson ($P < 0.005$) indices in the probiotic fed group (Table 4).

## Probiotic diet modified the microbial lineages and metabolic pathways in the hindgut

Linear discriminant analysis effect size (LEfSe) based on Wisconsin non-parametric *t*-test at 0.05 level of significance revealed three bacterial genus namely, *Illumatobacter*, *Clostridium* and *Cyanobium* to be enriched with *C. butyricum* supplementation in the marron (Fig. 5A). No significant ($P > 0.05$) enrichment of bacteria was observed in the control group. Piphillin and LEfSe based metabolic pathways extracted from KEGG database predicted an increased pathway activities for carbohydrate metabolism and absorption, amino acid metabolism, catabolism of fatty acid, degradation of toxic and synthetic compounds and
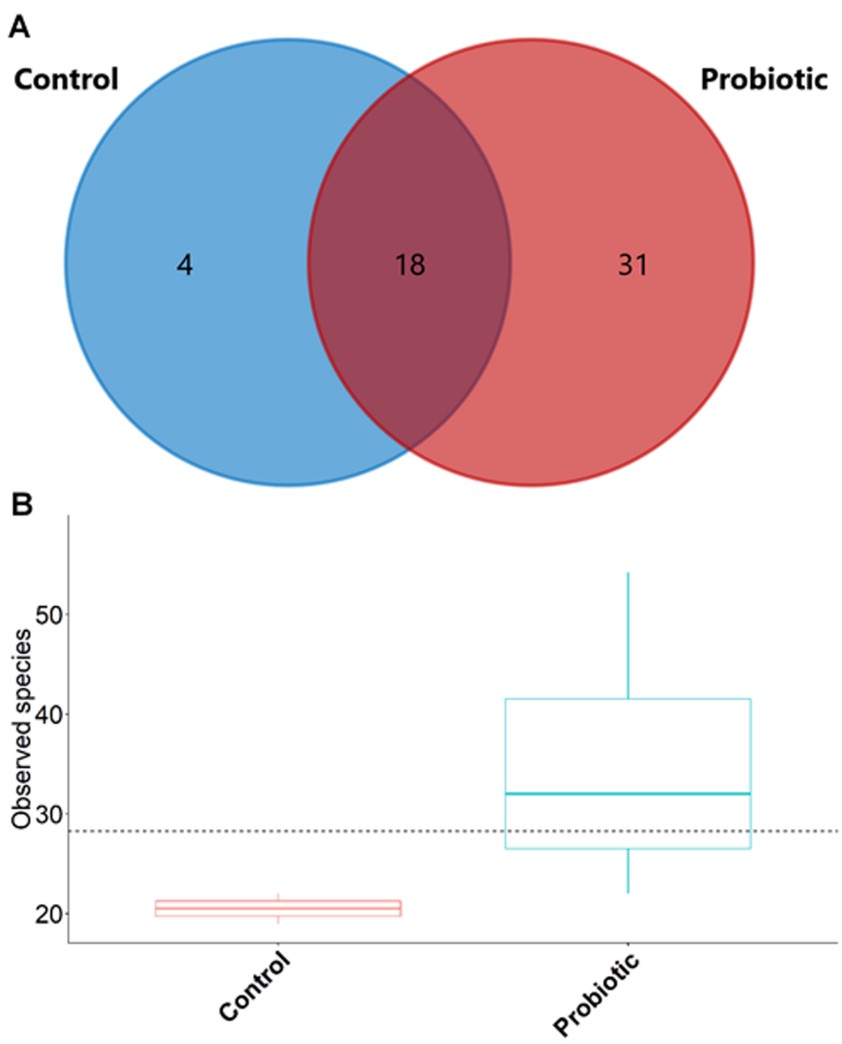

**Figure 3** (A) Venn diagram showing shared and unshared genus in two different groups; (B) box plots exhibiting the species richness in two different feeding conditions after trial.

biosynthesis of secondary metabolites in the probiotic fed group. In contrast, the analysis predicted significantly higher galactose, hormone and vitamin metabolism activities in the control group (Fig. 5B).

## Probiotic feed up-regulate the expression profile of immune responsive genes

The results of qRT-PCR for the selective genes associated with crayfish immune response post *in-vivo* challenge demonstrated significant ($P < 0.05$) up-regulation of pro-inflammatory cytokine (interleukin 17, IL17), anti-inflammatory cytokine (interleukin 10, IL10), cytosolic manganese superoxide dismutase (cytMnSOD) and prophenoloxidase (proPO). After 24 h of challenge, results revealed that the relative expression level of IL-17, IL-10, cytMnSOD and proPO were increased by 2.8, 2.5, 2.2 and 1.9 folds, respectively

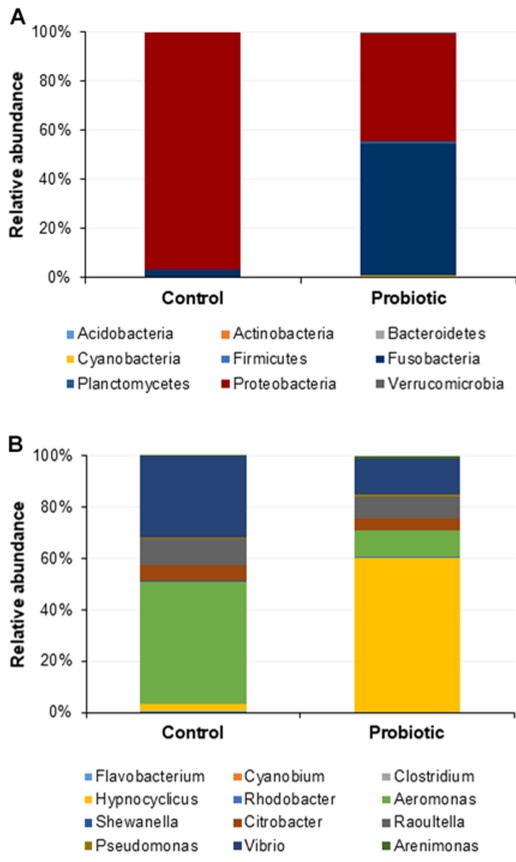

**Figure 4 Microbial communities in two different fed groups after trial; (A) relative abundance of bacterial OTUs at phylum level; (B) Relative abundance of bacterial OTUs at genus level.**

(Fig. 6). The expression level of cathepsin L (PcCTSL) and interleukin 8 (IL-8) genes were relatively static during post challenge test.

## Survival rate

At 24 and 96 h post-challenge, all marron injected with *V. mimicus* and PBS remained alive and started to respond to the feed given on day 5. The PBS injected marron, however, remained actively responsive to the feed given earlier (day 3) than bacteria injected marron. No signs of diseases were observed in any of the injected marron.

## Data availability

The raw FASTQ files are currently available in National Centre for Biotechnology Information (NCBI) BioProject under the accession number PRJNA515886.

## DISCUSSION

Probiotics as a dietary supplement have been in vogue for past 10 years for crayfish aquaculture to promote growth, disease resistance and stress tolerance (*Ambas, Fotedar & Buller, 2017*; *Ambas, Suriawan & Fotedar, 2013*; *Li, Tan & Mai, 2009*; *Zheng et al., 2017*).
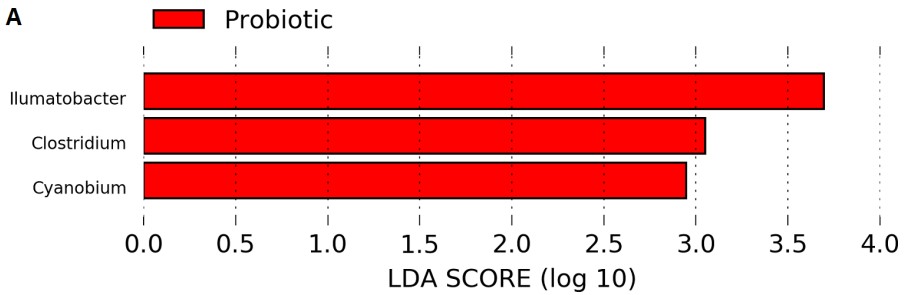

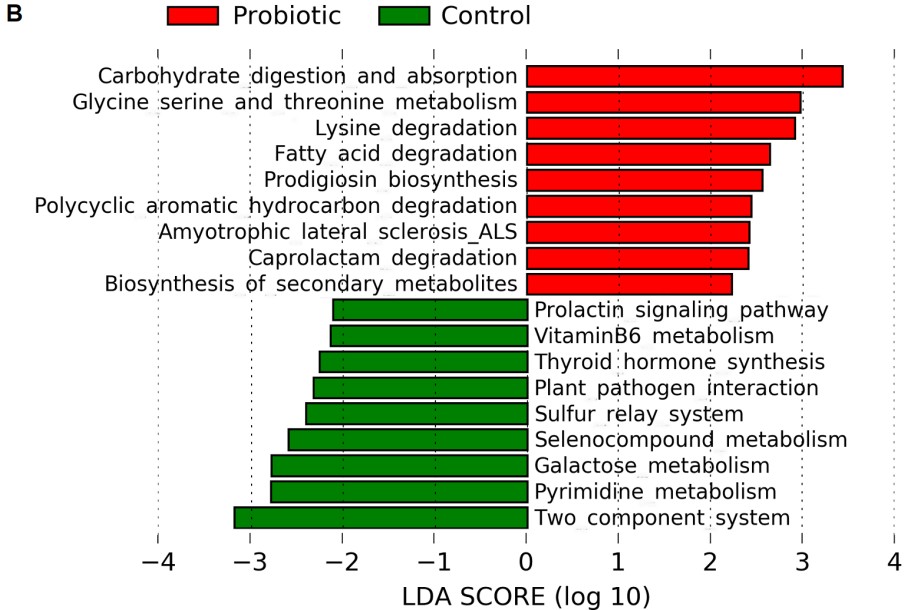

**Figure 5** **(A) Indicator bacterial genus in two different conditions (control and probiotic) with LDA value 2.0 or more; (B) predicted metabolic pathways in two different fed groups revealed from 16S rRNA sequence data using Piphillin and the LEfSe package.**

Most of the earlier researches on probiotics studies in aquatic animals were performed with the dietary inclusion of *Bacillus* and *Lactobacillus* species (*Ambas, Fotedar & Buller, 2017*; *Ambas, Suriawan & Fotedar, 2013*; *Kongnum & Hongpattarakere, 2012*; *Zheng et al., 2017*). However, a very few studies highlighted the importance of using *C. butyricum* as possible dietary probiotic candidate for crayfish (*Ramírez et al., 2017*). Results conducted with *C. butyricum* as dietary probiotic supplements demonstrated positive influence on growth and immune parameters of fish, chicken, piglet etc. (*Song et al., 2006*; *Pan et al., 2008*; *Gao et al., 2013*; *Abdel-Latif et al., 2018*; *Chen et al., 2018*). Reports also have shown that *C. butyricum* supplementation promote growth performance, body protein content of Pacific white leg shrimp, *Lotopenaeus vannamei* fed (*Duan et al., 2017*). Based on the findings of (*Duan et al., 2017*) study, we aimed to analyse the effects of *C. butyricum* on growth, microbial composition in gut and immune response of marron for the first time.

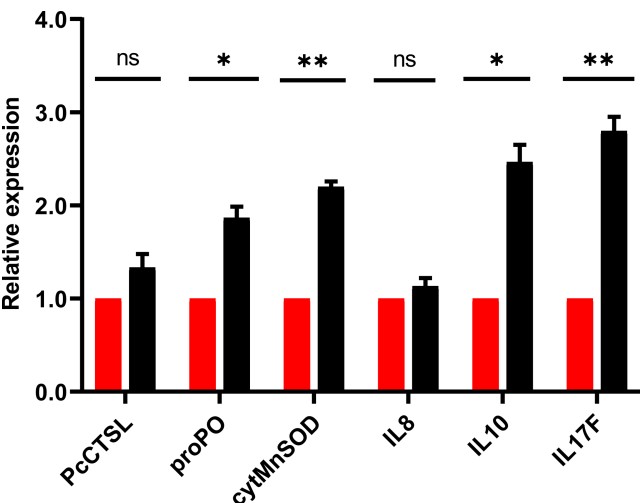

**Figure 6** Barplot showing the relative expression level of immune genes in marron intestine tissue after feeding trial. * Significantly different at $\alpha$-level of 0.05. **Significantly different at $\alpha$-level of 0.005.

Consistent with previous study by *Duan et al. (2017)*, we found significant ($P < 0.05$) improvement of growth and protein content in tail muscle of marron after 42 days of feeding trial. Besides this, *C. butyricum* supplementation in the diet also led to significantly increase the lysozyme and total haemocyte counts (THC) of haemolymph, indicating the possibility of using *C. butyricum* as a probiotic supplements in marron aquaculture.

Intestine of crayfish is regarded as the centre for digestion and absorption of nutrients whereas microbial communities present in the distal intestine play vital role in digestion and immunity (*Duan et al., 2017*). Besides the available core microbiota, enrichment with some bacterial communities facilitate to enhance the growth and immune performance of crayfish (*Hoseinifar et al., 2018*; *Li et al., 2018*). *C. butyricum* is a spore-forming butyric acid producing bacteria, common in animal gut that has many beneficial characteristics to be used as probiotic in poultry and aquaculture industry (*Pan et al., 2008*; *Van Immerseel et al., 2010*; *Takahashi et al., 2018*). Butyric acid has been widely used for animal nutrition owing to its positive effects on growth, intestinal digestion, absorption, metabolism and disease resistance (*Robles et al., 2013*; *Levy et al., 2015*; *Bourassa et al., 2016*). The dietary supplementation of *C. butyricum* in present study significantly (LDA > 2.0, $P < 0.05$) increased the abundance for *Clostridium* genus in the marron gut leading to enhanced growth and immune performance. In addition to *Clostridium*, the probiotic supplement also increased the richness of genus *Cyanobium* and *Illumatobacter*. Both of these bacteria identified from the gut and water demonstrated to have positive influence on remediation of water quality through the detoxification of pollutants and minimization of hypoxia (*Homonnay et al., 2012*; *Matsumoto et al., 2013*; *Wilhelm et al., 2014*; *Naoum, 2016*; *Yamaguchi et al., 2016*). The interactions among bacterial communities at phylum to species level are very complex, a single probiotic bacteria can influence the growth of whole microbial communities in the intestine of fish (*Stubbendieck et al., 2016*). A probiotic bacteria can influence a wide range of biological processes in fish and crayfish

including digestive enzymes, immune systems (phagocyte activity, clearance efficacy), pH of gastrointestinal tract, nutrient availability etc. (*Nayak, 2010*; *Sullam et al., 2012*). And hence, in a study, inclusion of single probiotic bacteria *Bacillus subtilis* in tilapia (*Oreochromis niloticus*) culture significantly ($P < 0.05$) modulated the diversity of six genus (*Giatsis et al., 2016*), a very similar findings to present study. At genus level, *C. butyricum* supplement significantly ($P < 0.05$) reduced the *Vibrio* and *Aeromonas* abundance in marron gut. *Vibrio* species has been associated with a number of diseases in crayfish and also considered as emerging pathogen for marron (*Ambas, Suriawan & Fotedar, 2013*; *Bean et al., 1998*; *Momtaz et al., 2019*). *Aeromonas hydrophilla* is widely reported as the pathogen for many crayfish species including *Pacifastacus leniusculus, Macrobrachium rosenbergii* (*Jiravanichpaisal et al., 2009*; *Abdolnabi et al., 2015*). Therefore, reduction of abundance for these two bacteria from the fish gut represents positive impact of probiotic diet on marron gut.

Although the use of probiotic bacteria for the growth and development of aquatic animals are widespread, however, the mode of action of these bacteria in crayfish gut is yet not clear. To investigate the effects of dietary *C. butyricum* supplementation on marron metabolic pathway, we applied simple, commonly used and straight-forward online tool, Piphillin (*Iwai et al., 2016*). The analysis predicted significantly higher activities for protein and energy metabolism, detoxification and secondary metabolites synthesis in the probiotic fed group. Furthermore, the positive effects of *C. byturicum* on crayfish, *Litopenaeus vannamei* growth, as reflected in terms of significant increase in enzymatic pathways activities for the metabolism of carbohydrate and protein were recorded (*Duan et al., 2017*). Consistent to this, our results predicted up-regulation of carbohydrate metabolism and absorption, amino acids (glycine, serine, threonine and lysine) metabolism and fatty acid metabolism. In addition, we also projected increased activities for biosynthesis of secondary metabolites including prodigiosin, degradation of toxic compounds including organic (polycyclic aromatic hydrocarbon, PAHs) and synthetic (caprolactam) waste. High PAHs concentration is harmful for aquatic life as they can persist for long time without being degraded by the natural system while microbial mediated biological degradation has been reported in the aquatic systems (*Noverita Dian, Aryanti & Nugraha, 2013*). The present study was designed to nurture marron without any filtration system, rather 30% water exchanged regularly. The untreated organic waste accumulated from the faecal and excess feed therefore mostly settled in the button of the tanks. The up-regulation of degradation pathway suggests bacterial decomposition of toxic organic waste might also be associated with higher growth in the probiotic fed marron.

The profiling of immune responsive genes post bacterial challenge test were performed to understand the role of probiotic feeding with the infection. Interleukins (IL) are the major class of cytokines associated with the immunity of crayfish (*Jiang et al., 2015*). Dietary supplementation of probiotic bacteria modulated the expression level of cytokine genes in fish (*Panigrahi, Viswanath & Satoh, 2011*; *Zokaeifar et al., 2012*; *Selim & Reda, 2015*). Present study also revealed significant upregulation of IL-17F and IL-10 expression in pathogen challenged marron. When pro-inflammatory cytokine (i.e., IL-17F) are up-regulated in challenged fish, the anti-inflammatory cytokine are also over-expressed

to prevent the damage from inflammation (*Miao et al., 2018*). Up-regulation of cytokine genes after feeding probiotic bacteria have been widely authenticated as positive influence to enhance immune performance of fish (*Panigrahi et al., 2007*; *Yang et al., 2014*; *Miao et al., 2018*). Two other significantly overexpressed genes, prophenoloxidase (proPO) and cytosolic manganese superoxide dismutase (cytMnSOD) reported to inhibit the growth of some crayfish pathogen i.e., shrimp white-spot virus (WSSV), *Vibrio* spp., and *Aeromonas hydrophilla* (*Liu et al., 2013*). Boosting immune performance by inhibition of *Vibrio* and *Aeromonas* after dietary administration of *C. butyricum* is in accordance with the finding of *Liu et al. (2013)*. Cathepsin L (PcCTSL) showed discrepancy in the expression level with previous study (*Dai et al., 2017*) which possibly due to differences in the composition of feed and shorter length of the previous study (two weeks).

## CONCLUSION

In summary, cellular and molecular based study revealed that supplementation of *C. butyricum* as a probiotic in feed improved growth performance, gut microbiota and immune response of marron. Therefore, *C. butyricum* could be utilised as potential probiotic supplement in the diet of marron. Further studies are required to reveal out the molecular mechanism of metabolic signalling pathways for improving the immune performance induced by *C. butyricum* in marron.

### Funding
This work was supported by the Australian Government, Department of Education and Training. The funders had no role in study design, data collection and analysis, decision to publish, or preparation of the manuscript.

### Grant Disclosures
The following grant information was disclosed by the authors:
Australian Government, Department of Education and Training.

### Competing Interests
The authors declare there are no competing interests.

### Author Contributions
- Md Javed Foysal conceived and designed the experiments, performed the experiments, analyzed the data, prepared figures and/or tables, authored or reviewed drafts of the paper, approved the final draft.
- Thi Thu Thuy Nguyen performed the experiments, approved the final draft.
- Md Reaz Chaklader and Muhammad A.B. Siddik prepared figures and/or tables, approved the final draft.
- Chin-Yen Tay analyzed the data, contributed reagents/materials/analysis tools, authored or reviewed drafts of the paper, approved the final draft.

- Ravi Fotedar conceived and designed the experiments, contributed reagents/materials/-analysis tools, authored or reviewed drafts of the paper, approved the final draft.
- Sanjay Kumar Gupta authored or reviewed drafts of the paper, approved the final draft.

## Data Availability

Data is available at National Centre for Biotechnology Information (NCBI), accession number: PRJNA515886.

## Supplemental Information

Supplemental information for this article can be found online at http://dx.doi.org/10.7717/peerj.7553#supplemental-information.

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
