# Peer review of "Marked variations in gut microbiota and some innate immune responses of fresh water crayfish, marron (Cherax cainii, Austin 2002) fed dietary supplementation of Clostridium butyricum"

_PeerJ, doi:10.7717/peerj.7553_

## Round 0.1 · original submission · Major Revisions

As you can see, your paper has received three reviewers. Some of them have raised a few issues that you need to consider before your paper could be accepted for publication. Please provide a point-by-point response to each one of the reviewers comments along with your revised manuscript.

·

Basic reporting

no comment

Experimental design

no comment

Validity of the findings

no comment

Additional comments

Dear Author,

The cumulative body of points indicate that a major revision is needed. There are lots of things that the authors should revised in the manuscript. Also, there are some questions and doubts which authors should address. find my comments and questions in details in the annotated pdf file.

Some comments:

First of all, after checking your gene expression results, I can say that your results would become more important in terms of scientific reality as it would be accepted as statistically significant if the expression rates of probiotic application exceeded 3 fold the rate of the control group. As far as I can see in the results, Probiotic has never increased or decreased the gene expressions 3 fold (at least) comparing to the control. Generally, statistical analyses is not performed if gene expression is not increased by at least 3-fold in scientific manuscript.

There is no clarity on the number of biological replicates used for different analysis, everywhere
it is mentioned “triplicate tanks”, which is not appropriate for parameters other than growth and FCR. Mention specifically the number of individual fishes (N=12) or pooled fish samples (N=4) used for immune parameters, gene expressions.... State this clearly in figure legends.
Only 4 fish samples were used for the immune parameters, gene expression analyses....of the fish. is it enough ?.

Why only intestine tissues were tested for immune genes expression? hepatopancreas was the most important immune organ for crayfish immunology, and therefore, author needs to validate related gene expression in this organ.

Growth, biochemical compositions, lysozyme, haemolymph osmolality, THC.. statistical informations should be added.

The lysozyme plates were incubated for 20 minutes, but that time can be adequate or not depending on the stopped of the enzyme to perform the lysis. Authors have to justify why they choose this time

The methods and reference of the method used to determine the LD50 in line 174 is not described.
Based on which you select this dosage for injection? LD50 ? Did the diseased crayfish have the signs of vibriosis?. Should be detailed.

Line 104. Culture conditions should be added.
Line 106. Moreover, there is no validation of the actual dietary levels of C. butyricum after dietary incorporation. In fact, the methodology used for dietary
incorporation is not convincing. There are questions related to the preparation of the C. butyricum in fish diet, top dressing of feed pellets without vacuum and final proximate composition of the experimental diets.

Table 1. references and Product size (bp) of the sequences should be added.

Reviewer 2 ·

Basic reporting

Yes, reporting is very thorough. Figures can be improved.

Experimental design

Yes; see below for more detailed comments.

Validity of the findings

Yes; see below for more detailed comments.

Additional comments

Marked variations in health status, gut microbiota and innate immune responses of marron (Cherax
cainii, Austin 2002) fed dietary supplementation of Clostridium butyricum (#34753)
by Foysal et al, PeerJ submission 34753.

The manuscript by Foysal et al describes the use of Clostridium butyricum as a probiotic supplement for marron, Cherax cainii, an increasingly important farmed crawfish in Australia.

This probiotic is a naturally found, acid producing bacteria, that resides in the gut of diverse animals and has been increasingly used as a probiotic in farmed organisms. Previously, the use of this probiotic has been shown to help reduce colonization by species of Vibrio and Aeromonas in fish, both of which are also known pathogens among crustaceans. The authors do admit, however, that in marron Vibrio are not yet known to cause disease (line 71).

Major points:

1.The authors demonstrate a correlation between molting and weight gain. Does age of the marrons factor in as well? Do the authors have some controls to take the age in consideration in these experiments.
2.In certain studies, the probiotic benefits of C. butyricum has been attributed to its ability to synthesize butyric acid (Duann et al 2017). Does this study explore the role of butyric acid in contributing to the probiotic effects of C. butyricum. Does modulation of fat synthesis by the probiotic lead to weight gain?
3.Authors demonstrate up-regulation of immune responsiveness induced by pre-exposure to probiotic followed by challenge with a potential pathogen. Can the results of these studies be supported further by survival assays?

line-by-line points:

line 58: define the type of acid produced.
line 71: remove ",however,"
line 76: change "with recently developed high-tech stuffs" to something else like, "with modern technologies such as"
line 93: this statement is not clear -- "in three randomly replicated tanks for each feeding treatment," which makes it unclear to the reader how samples were "pooled" as in line 151
line 112 (Dissolved Oxygen) - spelling mistake
line 223, Provide reference for growth techniques - It isn't mentioned in the said reference - Ambas, Fotedar 2016. Ramirez et al 2017 defines the same growth charachteristics with a different equation. Which one would be justified to use?
line 127: cacpodylate is misspelled; should read "cacodylate"
line 127: this is unclear to me. If the anticoagulant possesses fixative qualities, how do any of the cells or enzymes remain active for the assays?
line 135: please explain this assay a bit better; I believe that this solution of Micrococcus is a suspension of cell membrane components. Maybe explain to the reader how a turbidimetric assay works in a short statement.
line 155: NanoDrop does not really tell you much about quality (as a bioanalyzer would)
line 173: how were the 8 marron selected? Randomly?
line 173-74: a technical question, but why was the challenge bacteria not fed the way the probiotic was and instead injected? Under crowded conditions of farming, wouldn't the more natural exposure be an oral one, although certainly exposure can come via injury as well.
line 180: change "grinded" to "ground"
line 193 (onwards) for the Bioinformatics, when it is said that "pipelines" are used, were they used with default values? If so, please state it explicitly in the beginning of the section and then it will not be necessary to repeat it.
line 252-from here and discussion section: the authors should speculate on how it is possible that the use of a single probiotic species can somehow increase diversity in the gut? For example, enrichment of 3 specific genera as stated in line 270
line 318: it is suggested that probiotic supplementation increased abundance of the Clostridium genus. But I'm assuming it should only increase abundance of the specific probiotic? Unless the probiotic preparation is contaminated with other species? Otherwise, how then (as elsewhere in the paper), how does the presence of the probiotic help to increase overall microbial diversity in the gut?
line 336 (and section, line 268): please make it clear that Piphillin is a program that makes predictions on the presence of functional pathways based on the composition of the microbial communities present. And it does better of course with metagenomic data. For example, in line 337 is should be said "the analysis predicted significantly higher protein and energy metabolism" rather than saying that the analysis revealed...

Figures: overall the figures need to be improved. The font is barely legible. Overall, the legends can be improved.
Figure 1 (A) is unclear.
Figure 5: in A and B, it seems easier if the colors were consistent, for example, if probiotic is RED in A, then it should be RED in B.
Figure 6: it is unclear why the dendrogram is needed in this figure. What does the grouping tell us?

Reviewer 3 ·

Basic reporting

The language used is largely acceptable with clear technically correct text.
Introduction and discussion has been written nicely with relevant references and sound understanding of the subject.
The paper has been made more authentic by providing raw data set. The table, figures are appropriately described and depicted.
Overall the paper concludes with clear significant results.

Experimental design

The experiment has been designed with proper hypotheses to generate data to fill the knowledge gap. The experiments have been done conforming to the highest standards.
Methodology has described in detail for easy replication.

Validity of the findings

Clear results are obtained through robust data with excellent statistical analysis using latest bioinformatics tools available. Thus conclusions are well supported by results.

Additional comments

This is an excellent piece of work with lot of novelty in experimental design and execution. A very significant contribution has been made to support sustainable marron culture in Australia.

---

## Round 0.2 · accepted · Accept

Thank you for the thorough revision. Looking forward for your future submissions to PeerJ.

Reviewer 2 ·

Basic reporting

The authors have adequately addressed my concerns.

Experimental design

The experimental design is much clearer now.

Validity of the findings

The findings are robust.

Additional comments

Thank you for addressing these comments and concerns.